# Dosage individualization proposed for anti-gout medications among the patients with gout

**Binaya Sapkota**[1]*, **Suraj Chaudhary**[1], **Prakash Gurung**[2], **Anisha Humagain**[1], **Sujan Sapkota**[1]

**1** Nobel College Faculty of Health Sciences, Pokhara University, Kathmandu, Nepal, **2** Manmohan Memorial Medical College and Teaching Hospital, Kathmandu, Nepal

These authors contributed equally to this work.

* binaya@nobelcollege.edu.np

**Data Availability Statement:** All relevant data are within the manuscript and its Supporting information files.

## Abstract

### Background

The conventional one-size-fits-all approach has been criticized for almost all drugs used especially for chronic diseases, including gout. The present study was aimed to explore the need of individualization and optimization of the dose of anti-gout medications among gout patients.

### Methods

Cross-sectional study was carried out among 384 randomly selected new gout patients visiting two gout treatment centers at Lalitpur Metropolitan City, Nepal and who were taking antigout medications. Patients not taking anti-gout medications and not showing willingness to participate were excluded. The eGFR was calculated with the CKD Epidemiology Collaboration (CKD-EPI) creatinine equation (2009). Doses to be individualized were decided based on the Renal Drug Handbook and verified with the BNF 80. Data were analyzed via R 4.0.3 by applying the multinomial logistic regression to analyze statistical significance of risk with various predictors, and considering a p-value <0.05 statistically significant. Comorbidities were coded as per the ICD-11 coding and medicines were coded according to the WHO Guidelines for ATC classification and DDD assignment 2020.

### Results

The high risk of progression to CKD increased in the age range 54–63 and ≥84 years by 17.77 and 43.02 times, respectively. Also, high risk increased by 29.83 and 20.2 times for the overweight and the obese respectively. Aceclofenac 100mg was prescribed for maximum patients (30.5%). Need of dose individualization was realized in 30 patients, with maximum (7) in case of etoricoxib 90mg. Various glucocorticoids were prescribed for 36.9% patients, out of whom 3.8% required dose individualization and 15.9% patients with xanthine oxidase inhibitors, out of whom 1.3% required dose individualization.

**Funding:** The author(s) received no specific funding for this work.

**Competing interests:** The authors declare no potential conflicts of interest related to the present research and publication. Although the principal investigator is an Academic Editor of PLOS ONE, he had no influence over the independent editorial decision-making process of the journal, from its submission till the final decision. This does not alter our adherence to PLOS ONE policies on sharing data and materials.

## Conclusion

Thirty cases required dose individualization, which was although minimal but could have meaningful impact on the clinical success of the individual patient. Based on the recommendation on dose individualization, those patients could be optimized on their therapy on future follow ups.

## Introduction

Gout is an inflammatory arthropathy with a disorder of purine metabolism and is characterized by a sustained increase in blood uric acid, leading to urate crystal deposition and tissue damage [1, 2]. Clinically, it manifests as the recurrent acute and chronic arthritis, tophi, urolithiasis and renal disease. Gouty arthritis and tophi may lead to chronic disability, impair health-related quality of life (HRQoL), increase healthcare resources and reduce productivity [2–4]. Gout has prevalence of 1.14%, 1.7%, 2.49%, 2.7% and 4% in China, Australia, UK, New Zealand, and the USA respectively [1, 3, 5, 6]. Its prevalence is rising every year both in the developing and the developed countries due to various factors such as high body mass index (BMI) (as obesity increases endogenous uric acid production), and concomitant diseases (e.g., hypertension), consumption of purine-rich diets (e.g., red meat, high-fructose syrup or beverage, beer, seafood), and medications (e.g., diuretics, cyclosporine) [4, 6, 7–10]. Country-wise gout prevalence is understudied domain in Nepal like other developing countries [11], and district-wise prevalence was found to be 21.42% at Chitwan district, one of 76 districts in Nepal [12].

Gout has still remained mis-understood, mis- or under-diagnosed, and undertreated despite various recommendations and guidelines on its management [5, 8, 13, 14]. Guidelines for gout management usually help physicians select the most effective treatments and educate patients to promote adherence [13]. Acute gouty arthritis can be treated with colchicine, non-steroidal anti-inflammatory drugs (NSAIDs) (e.g., indometacin, naproxen), or glucocorticoids (oral and intra-articular injection) [8, 10, 15]. Xanthine oxidase inhibitors (XOIs) (e.g., allopurinol, febuxostat) are the first-line urate lowering therapies (ULTs) to prevent recurrent gouty arthritis whereas the uricosurics are the second-line medicines for those who cannot tolerate or show contraindications to XOIs [8]. Prophylaxis with colchicine should be administered followed by initiation of ULT (allopurinol or febuxostat). Patient education on risk factors for hyperuricemia and gout can improve therapeutic success as it is generally managed suboptimally [9]. Also, shared decision-making between the informed patients and the practitioners may improve its effective management [15].

The conventional one-size-fits-all approach has been criticized for almost all drugs used especially for chronic diseases, including gout. Therefore, individualization of drug therapy has been put forth to tackle the problem, as it helps optimize drug selection and dosing based on pathogenesis, mechanism of action (MOA) of drugs, and their dose exposure-response relationships. Dose individualization also helps optimize the efficacy of medicine by promoting its benefits and minimizing toxicity on individual patient. Dose can be individualized by various approaches such as selecting a medicine based on the effective serum concentrations, population pharmacokinetics, therapeutic drug monitoring (TDM), pharmacogenomics and other [16]. To date, very limited efforts have been undertaken in Nepal to individualize medications for chronic medication users. The present study was probably the first of its kind in

Nepal, aiming to explore the status of individualization and optimization of the dose of anti-gout medications among the gout patients.

## Methods

### Study design, area and population

Cross-sectional study was carried out among gout patients visiting two gout treatment centers (i.e., Gosainkunda Health Care Clinic, Satdobato, and Aarogya Health Home, Jawalakhel) at Lalitpur Metropolitan City, Nepal,who were taking antigout medications. Both of the study centers were privately owned, and patients were liable to pay for the medicine, laboratory, radiological and other health services out of their own pocket. Both of these centers were using manual prescriptions at the time of data collection. Every day 10 to 15 patients came to the first clinic, whereas 50 to 80 patients came to the latter, for gout treatment during the study period- from July to December 2019.

### Ethics approval

Approval for the research was obtained from the administrative department of both clinics, and ethics approval was received from Nobel College Institutional Review Committee (NIRC), Sinamangal, Kathmandu (Ref. No.: BPY IRC215/2019). The patients were verbally informed about the research objectives, and written informed consents were obtained from them prior to initiating the research. Their confidentiality was maintained throughout the research period (i.e., during both the pre- and post-completion phases of the research).

### Sampling and sample size

Simple random sampling (with lottery method) was followed for the data collection after sample frame was generated, and sample size was calculated based on Cochran's formula:

$$\text{Sample size (n)} = z^2 p(1-p)/d^2$$
$$= ((1.96)^2 \times 0.5 \times 0.5)/(0.05)^2 = 384$$

where, n = sample size; z = standard normal variate (1.96 for 95% confidence interval); p = population proportion; q = 1-p; d = absolute error or precision (0.05)

Since there were no exact data related to individualization and optimization of antigout medication dosage, we considered 50% population proportion while calculating the sample size for the current study.

### Inclusion and exclusion criteria

All patients taking anti-gout medications were enrolled for the research and their laboratory reports including creatinine value were considered. Those patients who were freshly prescribed with antigout medications and/or those taking the same since last six months (from the date of data collection) were enrolled. Patients not taking anti-gout medications and not showing willingness to participate were excluded. Also, patients taking antigout medications since beyond six months' time period were excluded from the study.

### Dosage individualization and optimization approaches

- Relevant patient-specific information such as demographic characteristics (e.g., age, gender, weight, and height), comorbidities, diet restrictions, and gout medications-related

information were obtained from the patients' prescriptions, once they finished consulting the physicians.

- The estimated glomerular filtration rate (eGFR) was calculated with the Chronic Kidney Disease Epidemiology Collaboration (CKD-EPI) creatinine equation (2009) using the National Kidney Foundation (NKF) GFR Calculator. The CKD-EPI creatinine equation was as follows: [17]

$$\text{eGFR } (\text{mL/min/1.73m}^2)$$
$$= 141 \text{ x } \min(S_{Cr}/\kappa, 1)^{\alpha} \text{x } \max(S_{Cr}/\kappa, 1)^{-1.209} \text{x } 0.993^{\text{Age}} \text{x } 1.018[\text{if female}]$$

where, $S_{Cr}$: standardized serum creatinine (mg/dL); $\kappa$ = 0.7 (females) or 0.9 (males); $\alpha$ = -0.329 (females) or -0.411 (males); min = minimum of $S_{Cr}/\kappa$ or 1; max = maximum of $S_{Cr}/\kappa$ or 1

- The GFR, albuminuria and risk categories were formed based on the criteria set by the NKF [17].

- Obesity status from the patients' respective BMI was computed based on the CDC guidelines [18].

- Finally, doses to be individualized were decided, based on the standard references of the Renal Drug Handbook 4th edition [19], and was verified with the British National Formulary (BNF) 80 [20]. Dose to be individualized was later discussed with the prescribing physicians.

## Data collection instrument, reliability and validity

We collected data in the self-developed, pre-piloted study tool, developed based on the extensive literature review [3, 4, 13]. The data collection sheet was divided into four parts to systematize the data collection process: the first part included patient demographics, second section covered patient-reported symptoms and physician-diagnosed disease profile along with comorbidities and dietary patterns, third part contained laboratory tests (e.g., hematological, biochemistry and serological), and the fourth part contained detailed treatment modalities with their individualization proposed. We performed pilot study among 10% of the projected sample size (i.e., 39 patients). Reliabilities of the tool and data collected on it were checked by Chronbach's alpha technique, whose value was 0.707. Face and content validity were checked based on the expert reviews and opinions on the questionnaire from five experts from the Department of Pharmaceutical Sciences, two experts from the Department of Medical Microbiology and one expert from the Department of Medical Laboratory Technology at Nobel College. Data collection from both of the study centers was done by four researchers, except the principal investigator, in collaboration with the practicing physicians. The need of the dose individualization was raised by the researcher pharmacists in collaboration with the physicians.

## Statistical analysis

We entered data into the IBM Statistical Package for the Social Sciences (SPSS) version 26 [21] and performed statistical testing via R programming 4.0.3 [22]. Categorical variables were presented with descriptive statistics. Multinomial logistic regression analysis was performed to analyze the statistical significance of the risk of progression to CKD with various predictors (e.g., age, gender, obesity, eGFR, duration of disease, and diet restrictions), considering a p-

value <0.05 statistically significant. Independent variables (i.e., predictors) were entered into the regression model, based on the evidence and stepwise pattern by following the principle of parsimony. Outcome variable was the risk of progression to CKD (with reference to the low risk), and such risk was determined by using the NKF-GFR Calculator [17]. Comorbidities of the patients were coded as per the International Statistical Classification of Diseases and Related Health Problems (ICD)-11 coding system [23], medicines were coded based on the WHO Guidelines for the anatomic/therapeutic/chemical (ATC) classification and defined daily dose (DDD) assignment 2020 23$^{rd}$ edition [24].

## Results

There were 384 enrollees in the study, 25.8% aged 44–53 years, 66.9%female, 50.5%participants with normal BMI (i.e., 18.5 to <25 kg/m$^2$), 70.6% patients with eGFR 76–125 mL/min/1.73m$^2$, 53.4%with normal or high GFR category (i.e., G1), 79.2% with duration of gout 0.2–10.1 years, and 67.7% on restriction of red meat, fish, alcohol and smoking, sour and spicy food, and suffering from the low risk of progression of CKD (Table 1).

The multinomial logistic regression analysis showed that the moderate risk of progression to CKD decreased with the advancing age. However, the risk of high progression increased in the age range 54–63 and ≥84 years by 17.77 and 43.02 times, respectively. The odds of the male suffering from the moderate and severe risks were 1/1.54 and 1/2.29e+07 respectively, showing less risk of gout among the males compared to the females. The high risk of gout increased by 29.83 and 20.2 times for the overweight and the obese patients, respectively. The high risk of CKD progression decreased with all the eGFR values from 26 to ≥176, whereas the moderate risk decreased for eGFR from 126 to ≥176. The moderate risk mainly increased by 20.47 and 34.29, provided that the duration of gout was 0.2–10.1 and 20–30.1 years, respectively (Table 2).

Aceclofenac 100 mg was prescribed for maximum patients (30.5%). The need of the dose individualization was realized for 30 patients, with maximum (7) in case of etoricoxib 90 mg (i.e., avoided if possible) (Table 3). There were 67.7% cases without associated comorbidities, followed by 9.6%patients with hyperthyroidism (S1 Table). Altogether, 57.5%patients were prescribed various NSAIDs, for whom 1.8% patients required dose individualization, all for etoricoxib 90 mg. Various intermediate-acting glucocorticoids were prescribed for 36.9% patients, out of whom 3.8% required dose individualization. Altogether, 15.9% patients were prescribed with XOIs, out of whom 1.3% required dose individualization (S2 Table).

## Discussion

Maximum patients (i.e., 69.3%) aged 34 to 63 years were suffering from the low risk of progression to CKD with gout. The high risk of progression increased in the age range 54–63 and ≥84 years by 17 and 43 times, respectively. The finding was in line with the various published findings that gout mainly affects men over the age of 40 years [6, 7, 14, 25–27]. Such slight variation might be due to the inclusion of all age groups in the present research. Since GFR is regarded as the best index of kidney function based on the data of serum creatinine, age, gender, race and/or body weight, and usually declines with the advanced age, gout may be more common among the aged [17].

Maximum female (66.9%) patients were suffering from the low risk of gout during the study period. However, other researches found that gout is three times more common among men compared to women [8, 14], the female accounting for only 5% of all gout patients, although in increasing trend [5, 10]. The incongruence in case of gender might be the reflection of the participation of more female patients during study period. Also, deposition of

**Table 1. Distribution of risk of progression of CKD with demographic and gout related characteristics of study population (n = 384).**

| Study variables | Risk of progression of CKD | | | Total (n,%) |
|---|---|---|---|---|
| | Low (n,%) | Moderate (n,%) | Very high (n,%) | |
| **Age (in years) (Mean±SD: 49.36±13.56)** | | | | |
| <= 3 | 0 | 1 (0.3) | - | 1 (0.3) |
| 4–13 | 1 (0.3) | - | - | 1 (0.3) |
| 14–23 | 10 (2.6) | - | - | 10 (2.6) |
| 24–33 | 31 (8.1) | - | - | 31 (8.1) |
| 34–43 | 84 (21.9) | 6 (1.6) | - | 90 (23.4) |
| 44–53 | 99 (25.8) | 6 (1.6) | - | 105 (27.3) |
| 54–63 | 83 (21.6) | 7 (1.8) | 1 (0.3) | 91 (23.7) |
| 64–73 | 32 (8.3) | 6 (1.6) | - | 38 (9.9) |
| 74–83 | 10 (2.6) | 5 (1.3) | - | 15 (3.9) |
| 84+ | 1 (0.3) | - | 1 (0.3) | 2 (0.5) |
| **Gender:** | | | | |
| Male | 94 (24.5) | 6 (1.6) | 1 (0.3) | 101 (26.3) |
| Female | 257 (66.9) | 25 (6.5) | 1 (0.3) | 283 (73.7) |
| **Obesity:** | | | | |
| Underweight: BMI <18.5 kg/m2 | 31 (8.1) | 4 (1) | - | 35 (9.1) |
| Normal: BMI 18.5 to <25 kg/m2 | 194 (50.5) | 17 (4.4) | - | 211 (54.9) |
| Overweight: BMI 25 to <30 kg/m2 | 106 (27.6) | 7 (1.8) | 1 (0.3) | 114 (29.7) |
| Obese: BMI ≥30 kg/m2 | 20 (5.2) | 3 (0.8) | 1 (0.3) | 24 (6.3) |
| **eGFR with the CKD-EPI creatinine equation (2009) (mL/min/1.73m2) (Mean±SD: 89.60±22.56)** | | | | |
| <= 25 | 0 | 0 | 1 (0.3) | 1 (0.3) |
| 26–75 | 67 (17.4) | 31 (8.1) | 1 (0.3) | 99 (25.8) |
| 76–125 | 271 (70.6) | - | - | 271 (70.6) |
| 126–175 | 12 (3.1) | - | - | 12 (3.1) |
| 176+ | 1 (0.3) | - | - | 1 (0.3) |
| **GFR category & CKD classification:** | | | | |
| G1: GFR ≥90 ml/min/1.73 m2 (Normal or high) | 205 (53.4) | - | - | 205 (53.4) |
| G2: GFR 60–89 ml/min/1.73 m2 (Mildly decreased) | 146 (38) | - | - | 146 (38) |
| G3a: GFR 45–59 ml/min/1.73 m2 (Mildly to moderately decreased) | - | 23 (6) | - | 23 (6) |
| G3b: GFR 30–44 ml/min/1.73 m2 (Moderately to severely decreased) | - | 8 (2.1) | - | 8 (2.1) |
| G4: GFR 15–29 ml/min/1.73 m2 (Severely decreased) | - | - | 2 (0.5) | 2 (0.5) |
| **Duration of disease (in years) (Mean±SD: 4.92±5.27)** | | | | |
| <= 0.1 | 7 (1.8) | - | - | 7 (1.8) |
| 0.2–10.1 | 304 (79.2) | 30 (7.8) | 2 (0.5) | 336 (87.5) |
| 10.2–20.1 | 35 (9.1) | - | - | 35 (9.1) |
| 20.2–30.1 | 3 (0.8) | 1 (0.3) | - | 4 (1) |
| 30.2+ | 2 (0.5) | - | - | 2 (0.5) |
| **Diet restriction:** | | | | |
| Beans, grains, sour and spicy foods, old pickle | 7 (1.8) | - | - | 7 (1.8) |
| Red meat, fish, alcohol and smoking, sour and spicy food | 260 (67.7) | 23 (6) | 2 (0.5) | 285 (74.2) |
| No restriction | 84 (21.9) | 8 (2.1) | - | 92 (24) |

eGFR: Estimated glomerular filtration rate; GFR: Glomerular filtration rate; MDRD: Modification of Diet in Renal Disease.

**Table 2.** Multinomial regression of risk of progression to CKD (with reference to low risk) with various predictors.

| Predictors | Moderate Risk | | | Very high Risk | | |
|---|---|---|---|---|---|---|
| | Beta (SE) | p-value | 95% CI for OR (lower & upper) | Beta (SE) | p-value | 95% CI for OR (lower & upper) |
| Intercept | -17.83 (92.87) | 0.84 | 1.79e-08 [1.58e-87, 2.04e+71] | -25.03 (131.23) | 0.84 | 1.33e-11 [2.61e-123, 6.83e+100] |
| **Age (in years):** | | | | | | |
| **4–13** | -72.17 (-) | - | 4.51e-32 [-, -] | -0.62 (2.46e-21) | <0.001 | 0.53 [0.53, 0.53] |
| **14–23** | -48.79 (1.17e-08) | <0.001 | 6.44e-22 [6.44e-22, 6.44e-22] | 0.94 (5.04e-12) | <0.001 | 2.57 [2.57, 2.57] |
| **24–33** | -75.41 (1.05e-08) | <0.001 | 1.76e-33 [1.76e-33, 1.76e-33] | -21.99 (1.94e-10) | <0.001 | 2.79e-10 [2.79e-10, 2.79e-10] |
| **34–43** | -33.70 (30.60) | 0.27 | 2.30e-15 [2.05e-41, 2.59e+11] | -2.86 (124.53) | 0.98 | 0.05 [5.6e-108, 5.78e+104] |
| **44–53** | -34.37 (30.60) | 0.26 | 1.17e-15 [1.04e-41, 1.31e+11] | -36.48 (2.03e-16) | <0.001 | 1.43e-16 [1.43e-16, 1.43e-16] |
| **54–63** | -34.38 (30.60) | 0.26 | 1.16e-15 [1.04e-41, 1.3e+11] | 17.77 (189.06) | 0.92 | 5.26e+07 [6.13e-154, 4.52e+168] |
| **64–73** | -34.45 (30.60) | 0.26 | 1.08e-15 [9.66e-42, 1.22e+11] | -22.82 (4.37e-09) | <0.001 | 1.22e-10 [1.22e-10, 1.22e-10] |
| **74–83** | -32.81 (30.60) | 0.28 | 5.59e-15 [4.94e-41, 6.31e+11] | -2.56 (2.93) | 0.38 | 0.07 [0.0002, 24.33] |
| **84+** | -16.34 (8850050) | 0.83 | 7.97e-08 [6.08e-75, 1.04e+60] | 43.02 (0.0003) | <0.001 | 4.84e+18 [4.84e+18, 4.84e+18] |
| **Gender** (Female): | 0.43 (0.68) | 0.52 | 1.54 [0.4, 5.93] | 16.95 (132.61) | 0.89 | 2.29e+07 [3.04e-106, 1.73e+120] |
| **Obesity** | | | | | | |
| **Normal** | 0.12 (0.85) | 0.88 | 1.13 [0.21, 6.004] | 17.07 (276.09) | 0.95 | 2.61e+07 [2.51e-228, 2.71e+242] |
| **Overweight** | -0.06 (0.90) | 0.94 | 0.93 [0.15, 5.53] | 29.83 (154.93) | 0.84 | 9.02e+12 [1.18e-119, 6.84e+144] |
| **Obese** | 0.51 (1.11) | 0.64 | 1.67 [0.18, 14.84] | 20.20 (196.36) | 0.91 | 5.95e+08 [4.24e-159, 8.33e+175] |
| **Class 1 obesity** | 0 (7.01e-14) | 1 | 1 [1, 1] | 0 (0) | - | 1 [1, 1] |
| **Class 2 obesity** | 0 (-) | - | 1 [-, -] | 0 (0) | - | 1 [1, 1] |
| **Class 3 obesity** | 0 (7.99e-14) | 1 | 1 [1, 1] | 0 (0) | - | 1 [1, 1] |
| **eGFR_CKD:** | | | | | | |
| **26–75** | 30.44 (34.39) | 0.37 | 1.67e+13 [8.82e-17, 3.16e+42] | -28.04 (228.53) | 0.90 | 6.59e-13 [1.96e-207, 2.21e+182] |
| **76–125** | 3.59 (98.66) | 0.97 | 36.41 [3.8e-83, 3.48e+85] | -55.22 (370.94) | 0.88 | 1.03e-24 [0, 5.77e+291] |
| **126–175** | -20.51 (2.09e-16) | <0.001 | 1.23e-09 [1.23e-09, 1.23e-09] | -45.57 (4.28e-10) | <0.001 | 1.60e-20 [1.6e-20, 1.6e-20] |
| **176+** | -24.31 (1.16e-18) | <0.001 | 2.74e-11 [2.74e-11, 2.74e-11] | -3.74 (124.48) | 0.97 | 0.02 [2.56e-108, 2.18e+104] |
| **Duration (in years):** | | | | | | |
| **0.2–10.1** | 20.47 (129.94) | 0.87 | 7.79e+08 [1.92e-102, 3.16e+119] | -12.52 (247.32) | 0.95 | 3.62e-06 [1.08e-216, 1.2e+205] |
| **10.2–20.1** | 2.47 (1.19) | 0.03 | 11.86 [1.14, 122.50] | 0.41 (373.86) | 0.99 | 1.52 [8.8e-319, ∞] |
| **20.2–30.1** | 34.29 (217.89) | 0.87 | 7.84e+14 [2.62e-171, 2.34e+200] | 1.86 (3.93e-11) | <0.001 | 6.44 [6.44, 6.44] |
| **30.2+** | -23.07 (2.10e-12) | <0.001 | 9.52e-11 [9.52e-11, 9.52e-11] | -6.62 (0.001) | <0.001 | 0.001 [0.001, 0.001] |

Model: P(Risk of progression to CKD) = $1/\{1 + e^{-(a + b1Age + b2Gender + b3Obesity + b4eGFR\_CKD + b5Duration\ of\ disease)}\}$.

Body weight: Normal: BMI 18.5 to <25 kg/m2; Overweight: BMI 25 to <30 kg/m2; Obese: BMI 30 kg/m2; Class 1 obesity: BMI 30 to <35 kg/m2; Class 2 obesity: BMI 35 to <40 kg/m2; Class 3 obesity (extreme or severe): BMI 40 kg/m2.

monosodium urate crystals in any of the age groups in the gender non-specific pattern may lead to chronic arthritis, tophi, urolithiasis and renal disease [26]. Further, gout may be misdiagnosed or subject to the delayed diagnosis due to insidious attack among the women with the protective effects of estrogen during premenopausal phase, and the rising incidence after menopause, peaking at ≥80 years among the women [5, 10].

Overweight men with BMI >27.5 kg/m² usually have up to 16 times higher risk for gout than those with normal BMI with <20 kg/m² [9]. The present research also showed that the high risk of gout increased by 29 and 20 times for the overweight and the obese patients, respectively. This might be aggravated by comorbidities as 67.7% cases without associated comorbidities, followed by 9.6% with hyperthyroidism seemed to have the risk of gout. Other researches also concluded that gout may appear with various co-morbidities including obesity,

**Table 3. Medicines used with the status of dose individualization (n = 384).**

| SN | Medicines | Therapeutic category | ATC Classification* | Not used (n, %) | Used (n,%) | No Dose adjustment required (n,%)** | Dose individualization required (n,%) (Total: 30, 8%)** | Individualized dose |
|---|---|---|---|---|---|---|---|---|
| 1 | Aceclofenac 100 mg | Preferential COX-2 inhibitor | MO1A | 267 (69.5) | 117 (30.5) | 117 (30.5) | - | NA |
| 2 | Allopurinol 100 mg | XOI | M04AA | 383 (99.7) | 1 (0.3) | 1 (0.3) | - | NA |
| 3 | Colchicine 500 mcg BD | Alkaloid | M04AC01 | 344 (89.6) | 40 (10.4) | 36 (9.4) | Reduce dose or dosage interval by 50%: 4 (1) | 250 mcg |
| 4 | Etoricoxib 60 mg | NSAIDs: COX-2 inhibitor | M01AH05 | 383 (99.7) | 1 (0.3) | 1 (0.3) | - | NA |
| 5 | Etoricoxib 90 mg | | | 292 (76) | 92 (24) | 85 (22.1) | Dose as in normal renal function, but avoid if possible: 7 (1.8) | Avoid if possible |
| 6 | Etoricoxib 90/60 mg | | | 383 (99.7) | 1 (0.3) | 1 (0.3) | - | NA |
| 7 | Febuxostat 40 mg | XOI | M04AA | 331 (86.2) | 53 (13.8) | 49 (12.8) | Start with 40 mg and monitor closely: 1 (0.3); Dose as in normal renal function: 3(0.8) | 40 mg with close monitoring |
| 8 | Febuxostat 80 mg | | | 377 (98.2) | 7 (1.8) | 6 (1.6) | Dose as in normal renal function: 1(0.3) | Dose as in normal renal function |
| 9 | Indometacin 25 mg | NSAIDs: Indole acetic acid derivative | M01AB01 | 383 (99.7) | 1 (0.3) | 1 (0.3) | - | NA |
| 10 | Indometacin 50 mg | | | 378 (98.4) | 6 (1.6) | 6 (1.6) | - | NA |
| 11 | Methylprednisolone inj. 80 mg | Glucocorticoid: Intermediate acting | H02AB04 | 355 (92.4) | 29 (7.6) | 28 (7.3) | Dose as in normal renal function: 1(0.3) | Dose as in normal renal function |
| 12 | Naproxen 250 mg | NSAIDs: Propionlc acid derivatives | M01AE02 | 382 (99.5) | 2 (0.5) | 2 (0.5) | - | NA |
| 13 | Prednisolone 2.5 mg | Glucocorticoid: Intermediate acting | H02AB06 | 352 (91.7) | 32 (8.3) | 29 (7.6) | Dose as in normal renal function: 3(0.8) | Dose as in normal renal function |
| 14 | Prednisolone 5 mg | | | 354 (92.2) | 30 (7.8) | 27 (7) | Dose as in normal renal function: 3(0.8) | Dose as in normal renal function |
| 15 | Prednisolone 10 mg | | | 372 (96.9) | 12 (3.1) | 10 (2.6) | Dose as in normal renal function: 2(0.5) | |
| 16 | Prednisolone 15 mg | | | 379 (98.7) | 5 (1.3) | 3 (0.8) | Dose as in normal renal function: 2(0.5) | |
| 17 | Prednisolone 20 mg | | | 374 (97.4) | 10 (2.6) | 9 (2.3) | Dose as in normal renal function: 1(0.3) | |
| 18 | Prednisolone 25 mg | | | 382 (99.5) | 2 (0.5) | 2 (0.5) | - | NA |
| 19 | Prednisolone 30 mg | | | 369 (96.1) | 15 (3.9) | 14 (3.6) | Dose as in normal renal function: 1(0.3) | Dose as in normal renal function |
| 20 | Prednisolone 30/20/10 | | | 383 (99.7) | 1 (0.3) | 1 (0.3) | - | NA |
| 21 | Prednisolone 20/15/10 | | | 383 (99.7) | 1 (0.3) | 1 (0.3) | - | NA |
| 22 | Prednisolone 15/10/5/ 2.5 mg | | | 383 (99.7) | 1 (0.3) | 1 (0.3) | - | NA |
| 23 | Prednisolone 15/10/5 | | | 383 (99.7) | 1 (0.3) | 1 (0.3) | - | NA |
| 24 | Prednisolone 5/2.5 | | | 383 (99.7) | 1 (0.3) | - | Dose as in normal renal function: 1(0.3) | Dose as in normal renal function |

(*Continued*)

**Table 3.** (*Continued*)

| SN | Medicines | Therapeutic category | ATC Classification* | Not used (n, %) | Used (n,%) | No Dose adjustment required (n,%)** | Dose individualization required (n,%) (Total: 30, 8%)** | Individualized dose |
|----|-----------|---------------------|---------------------|-----------------|------------|-------------------------------------|----------------------------------------------------------|---------------------|
| 25 | Tapentadol 50 mg | Atypical opioid | D06007 | 381 (99.2) | 3 (0.8) | 3 (0.8) | - | NA |
| 26 | Triamcinolone 10 mg inj. | Glucocorticoid: Intermediate acting | D07XB02 | 383 (99.7) | 1 (0.3) | 1 (0.3) | - | NA |

NA: not aplicable (i.e., as usual dose).

*WHO Guidelines for ATC classification and DDD assignment 2020.

** The Renal Drug Handbook: The ultimate prescribing guide for renal practitioners 4th edition (2014).

dyslipidemia, cardiovascular disease (e.g., diabetes mellitus, hypertension), CKD, hypothyroidism, anemia, and chronic pulmonary diseases [2–4].

The eGFR is a reliable diagnostic test to detect and manage kidney disease, plan for dialysis, evaluate the success of renal transplantation, if any, and to optimize medications dosing, and it gets declined with the advancing age [17]. The high risk of CKD progression decreased with all the eGFR values from 26 to ≥176, whereas the moderate risk decreased with the eGFR from 126 to ≥176. The moderate risk increased by 20.47 and 34.29 times, provided that the duration of gout was 0.2–10.1 and 20–30.1 years, respectively indicating that the more chronic the disease becomes, the more deteriorating effects might appear.

The research found that 67.7% low-risk patients were suggested to refrain from consuming red meat, fish, alcohol and smoking, and spicy food to prevent from their progression to CKD. However, Wise (2005) reported that the dietary restrictions alone may not have major impacts on the elderly gout patients, and even the purine-free diets may have low effect on maintaining the target serum urate [28]. However, treatment approaches are the same for both the men and the women with gout, such as avoiding risk factors (e.g., alcohol, diuretics), maintaining normal serum glucose levels, blood pressure, body weight and BMI, and consuming less purine-rich diets (e.g., red meat, fructose, seafood) [10].

## Dose individualization approaches

Acute gout flare can be treated, and even the prophylaxis can be provided with colchicine, NSAIDs, and corticosteroids [9], and 10.4%, 57.5% and 36.9% patients were prescribed with these medicines in the present research, either alone or combined. Four of the colchicine users required dose adjustment, and its lower starting doses (i.e., maximum of 3 tablets of 500 mcg in the first 24 hour) are preferred to avoid untoward effects without compromising efficacy [29]. Colchicine 500 mcg twice daily, and 500 mcg/day (i.e., reduced frequency) can be used by the patients with normal renal function and renal impairment, respectively [30], which was the case in the present research too. Renal function status should be assessed before initiating colchicine or NSAIDs [26], and colchicine is better avoided by the elderly, as this is poorly tolerated among that group [31]. Colchicine or NSAID prophylaxis should be given when antihyperuricemic therapy is initiated, and should be continued for a prolonged period till the patient is free from the gout attacks [32].

Seven of the NSAIDs users required dose adjustment, and such dose individualization was proposed for etoricoxib 90 mg (i.e., dose as in the normal renal function, but avoided if possible). The NSAIDs with the short half-lives (<6 hr) (e.g., diclofenac and ketoprofen) are preferred in the acute gout patients with the concurrent renal failure, uncontrolled hypertension or cardiac failure [29, 31], and these should be initiated at their maximum dose in case of acute

gout, rather than titrating upwards from the lower dose [14, 29]. For e.g., indometacin 150–200 mg/day, naproxen 1000 mg/day, diclofenac 150 mg/day are used for 4–8 days and then gradually tapered till symptoms get resolved [29]. But the case was not evident in the present study as indometacin was found to be prescribed in 25 and 50 mg doses, and naproxen in 250 mg dose, with no diclofenac being prescribed. The NSAIDs are usually contraindicated in the elderly, and the low-dose colchicine (500 mcg/day) or prednisone 5–7.5 mg/day may be used for the first 6 months after initiating allopurinol [33].

Fourteen of the intermediate-acting glucocorticoids users required dose individualization in the present study. Prednisone can be started at a dose of 20–40 mg/day, and gradually tapered and discontinued over 10–14 days. However, corticosteroids should be cautiously used by patients with diabetes mellitus, hypertension, congestive heart failure (CHF), coronary artery disease (CAD) and severe infections [29].

Sixty one patients were prescribed with XOIs, out of whom 5 required dose individualization, all for febuxostat. The first-line ULT (i.e., allopurinol) was found to be prescribed for only a single patient, and the second-line (i.e., febuxostat) for 60 patients. The second-line ULT isused if the first-line is not tolerated or the response is inadequate [1, 2] or when uricosurics are contraindicated (as in case of stage 3 CKD, urolithiasis, or uric acid overproduction) [34]. Febuxostat might have been preferred to allopurinol in the present study due to its few drug interactions compared to the latter, and its minimal propensity to cause allopurinol hypersensitivity syndrome (AHS) [30, 35–37]. Also, it has added benefit that its dose adjustment is usually not necessary in case of mild-to-moderate renal or liver insufficiency or the advanced age [8, 37–39].

Febuxostat 40mg was prescribed for 53 patients, whereas its 80mg dose for 7 patients in the present research. Majority of the febuxostat users were started with a dose of 40mg/day as expected, but 80mg could be used in case the serum uric acid (sUA) levels were <6mg/dL even two weeks after therapy is initiated [38]. Febuxostat 40mg/day has been found to be equivalently efficacious to allopurinol 300mg/day [30, 39, 40] and its 80mg/day dose has superior urate-lowering efficacy compared to allopurinol 300mg/day [37, 40]. Its dose can be increased to 120mg/d after 4 weeks to reach the therapeutic target for sUA levels [26, 38], but none of the patients reached up to that level in the present study. Patients should be educated on the lifestyle modifications (e.g., exercise, consumption of the low purine diets) along with the medications to reduce the raised serum urate levels. Weight reductions with exercise show a positive impact on the urate reduction [32]. The study findings may help the practitioners select the optimum regimen of anti-gout therapy for the individual patient based on the dosage individualization proposed.

## Strengths and limitations of the study

The report of the individualization status, based on the Renal Drug Handbook was provided to the concerned physicians so that they might be aware of the need of the same for the patients in their scheduled follow ups. This might help the physicians optimize the therapeutic success, once they implement it, and the patients adhere to it. However, the present study was not free from some of the limitations including:

- The study was limited to only two gout and rheumatic diseases treatment centers, which might not represent the whole gout patients in the nation.

- Medication non-adherence was not measured in the present study.

- Large controlled clinical trials could be conducted based on the foundation of the present study.

## Conclusions

Majority of patients (57.5%) were prescribed with NSAIDs and 1.8% of whom required dose individualization. Thirty cases required dose individualization, which was although minimal but could have meaningful impact on the clinical success of the individual patient. The dose individualization was felt necessary for those patients taking colchicine 500mcg, etoricoxib 90mg, febuxostat 40mg and 80mg, methylprednisolone inj. 80mg, prednisolone 2.5mg, 5mg, 15mg 20mg and 30mg, and prednisolone 5/2.5 mg. Based on the dose individualization recommendation of the present study, those patients could be optimized therapeutically on their future follow ups, providing optimal benefits to them. Also, future randomized controlled trials may be based on the findings of the present study to derive a more conclusive evidence base for gout management.

## Supporting information

**S1 Table. Comorbidities of the patients [23].**
(DOCX)

**S2 Table. Therapeutic category-wise medication use.**
(DOCX)

## Acknowledgments

The authors are grateful to all the respondents for their valuable time and active cooperation throughout the research. They are also grateful to the physicians and the administrators at the two study clinics, who provided their intellectual and managerial support throughout the research.

## Author Contributions

**Conceptualization:** Binaya Sapkota.

**Data curation:** Suraj Chaudhary, Prakash Gurung, Anisha Humagain, Sujan Sapkota.

**Formal analysis:** Binaya Sapkota.

**Investigation:** Binaya Sapkota, Sujan Sapkota.

**Methodology:** Binaya Sapkota.

**Project administration:** Binaya Sapkota, Suraj Chaudhary, Prakash Gurung, Anisha Humagain, Sujan Sapkota.

**Resources:** Binaya Sapkota, Suraj Chaudhary, Prakash Gurung, Anisha Humagain.

**Software:** Binaya Sapkota.

**Supervision:** Binaya Sapkota.

**Validation:** Binaya Sapkota, Suraj Chaudhary, Prakash Gurung, Anisha Humagain, Sujan Sapkota.

**Visualization:** Binaya Sapkota, Prakash Gurung, Anisha Humagain, Sujan Sapkota.

**Writing – original draft:** Binaya Sapkota, Suraj Chaudhary, Prakash Gurung, Anisha Humagain, Sujan Sapkota.

**Writing – review & editing:** Binaya Sapkota, Suraj Chaudhary, Prakash Gurung, Anisha Humagain, Sujan Sapkota.

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
