## [Decision Letter · Decision Letter 0]

23 Mar 2021

PONE-D-20-38874

Dosage individualization proposed for anti-gout medications among the patients with gout: A bicentric study in Nepal

PLOS ONE

Dear Dr. Sapkota,

Thank you for submitting your manuscript to PLOS ONE. After careful consideration, we feel that it has merit but does not fully meet PLOS ONE’s publication criteria as it currently stands. Therefore, we invite you to submit a revised version of the manuscript that addresses the points raised during the review process.

We look forward to receiving your revised manuscript.

Kind regards,

Tauqeer Hussain Mallhi, Ph.D

Academic Editor

PLOS ONE

Journal Requirements:

Additional Editor Comments (if provided):

Dear Authors, thank you for submitting in Plos One. Your manuscript has been assessed by relevant experts from the field. They found manuscript interesting but raised some concerns in methodology (study population, statistical analysis) and interpretation of results. It is requested to please consider the comments of reviewers.

Reviewers' comments:

Reviewer's Responses to Questions

**Comments to the Author**

1. Is the manuscript technically sound, and do the data support the conclusions?

Reviewer #1: Partly

Reviewer #2: Partly

Reviewer #3: Partly

Reviewer #4: No

Reviewer #5: Yes

Reviewer #6: Yes

2. Has the statistical analysis been performed appropriately and rigorously? 

Reviewer #1: No

Reviewer #2: Yes

Reviewer #3: Yes

Reviewer #4: No

Reviewer #5: Yes

Reviewer #6: Yes

3. Have the authors made all data underlying the findings in their manuscript fully available?

Reviewer #1: Yes

Reviewer #2: Yes

Reviewer #3: Yes

Reviewer #4: No

Reviewer #5: No

Reviewer #6: Yes

4. Is the manuscript presented in an intelligible fashion and written in standard English?

Reviewer #1: No

Reviewer #2: No

Reviewer #3: No

Reviewer #4: Yes

Reviewer #5: Yes

Reviewer #6: Yes

5. Review Comments to the Author

Reviewer #1: The current bi-centric cross-sectional study conduct by Sapkota et al., has evaluated the dosage individualization proposed for anti-gout medications in the patients suffering from at two hospitals in Nepal. The authors have found that a total of 8% of the study participants needed dose individualization. This is an interesting topic, but the way the manuscript is presented makes it difficult for the readers to understand the findings of the study. Some of my comments to the authors are as follows:

Major comments

The authors have written down that “present study was aimed to individualize and optimize the dose of anti-gout medications among gout patients”. However their main findings are about frequency of patients who needed dose individualization. The aim of the study should redrafted.

The authors have used multinomial regression to find factors associated with the risk of progression of CKD in the study participants. First, this is not the major aim of the study and one gets confused that what is the need of this analysis in the current manuscript. Second, the results of multinomial regression have not been presented in the abstract. Third, progression or deterioration of a disease cannot be evaluated by a cross-sectional study design. What the authors have presented in the Results section of the main document are just the factors which have significant association with advanced stages of kidney diseases. It would have been good, if the authors have limited the results and discussion to the primary objective of the study. i.e. frequency of patients who needed dose individualization and the factors associated with it.

What criteria has been used for entering the independent variables into multinomial regression? Why 50% population proportion has been used in calculating the sample size for the current study? The authors should also give a bit detail about their random selection of the study subject, what type of random selection they have employed? How many patients or their records were approached, how many of them have been not included in the study and why?.

The value of “z” in the sample size calculation formula has been mistakenly written as 0.96, it should be written correct as 1.96.

In the methods section of the paper it has been written as “…..study was carried out among new gout patients visiting two gout treatment centers”, whereas in the Inclusion, exclusion part of the paper it has been written as “……All patients taking anti-gout medications along with laboratory reports consisting of creatinine value were included for the research”. As the authors have included established gout patients who were taking anti-gout medicines, the authors should make it clear that they mean by “new gout patients”

In the results sections the authors have mentioned that “….The multinomial logistic regression analysis showed that moderate risk of progression to CKD decreased with advancing age but that of high risk increased in the age range 54-63 and ≥84 years by 17.77 and 43.02 times, respectively”. What the authors meant by moderate risk and high risk? What criteria has been used for categorizing the risk of renal disease progression as moderate and high?

Overall the discussion is too vague and does not revolve around the main findings of the study.

There are many typo and grammatical mistakes which need correction.

Reviewer #2: Dear Editor

The following are my observations on the article “Dosage individualization proposed for anti-gout medications among the patients with gout: a bicentric study in Nepal”.

1. The inclusion and the exclusion criteria to be more specific:

Example: The patients taking anti gout medications are recruited in the study, the authors does not mention, how long the patients on medications can be recruited. Based on the duration of treatment, the outcome of the assessment varies. Similarly, the exclusion criteria need to be specific.

2. In the discussion section, most of the data mentions about the results. Discussion should be re-written.

With regards

Dr. Roopa B S

Reviewer #3: Comment 1: Typographical error was observed while calculating sample size.

Comment 2: Grammatical errors were observed in exclusion criteria, it is giving different meaning, please rewrite the sentence.

Comment 3: Abbreviations should be defined at first mention and used consistently thereafter.

Comment 4: The word country-wide supposed to be country-wise in line 1 and 2 at page 3. Kindly check it.

Comment 5: Please re-write the line number 3 and 4 at page number 3.

Comment 6: In data section, can you elaborate the tools used for assessment, or the questionnaire used and the value of Cronbach analysis

Comment 7: In the conclusion section, can you elaborate that which of the 30 individuals require the individualization.

Overall, this manuscript needs major revision including write-up and supportive evidence.

Reviewer #4: Comments

1. Please describe SRS procedure.

2. Researcher perform any pilot study? If yes then please describe. If not how it could be fulfill face validation? Which is required for any developed questionnaire.

3. Better to report the content validity results with CVR and CVI.

4. Outcome variable is not describe properly.

5. What was the variable selection procedure for Multinomial regression model?

6. Imaginary results observed from Multinomial regression especially from OR and CI. Due to inappropriate grouping of predictors (lot of groups included zero frequency) as well as inappropriate reference category inflation happened on OR and CI.

7. Suggesting to reanalyze data after performing proper categorization among predictors with bivariate analysis to select predictors for multiple model.

Reviewer #5: Dosage individualization proposed for anti-gout medications among the patients with gout: A bicentric study in Nepal

PONE-D-20-38874

Thank you very much for the opportunity to review this manuscript. I invite authors to resubmit your manuscript after addressing all comments. Please carefully consider all issues I mentioned, outline every change made point by point and send us 2 documents: revised manuscript with track changes and author responses to the reviewer comments. Thanks

Abstract:

Objective: Was the objective to optimize the therapy or prove the necessity of the dose optimization to minimize the predicted progression to chronic kidney disease (CKD) ? As the conclusion shows the need (thirty (8%) cases required dose individualization) rather than outcome of the individualization.

Methods: Bicentric cross-sectional” The term bicentric can be deleted since it is not part of the study design (cross-sectional) while it is part of the study setting (in 2 centers).

Not showing unwillingness. Please do not use double negative, so reword to either “not showing willingness” or showing unwillingness.

I am not sure how cross-sectional design can prove medicine individualization? I mean without prospective follow-up of the cases how can the authors prove the effectiveness of the individualized regimen?

Methods

Study design, study area and study population

Please specify whether those 2 outpatient clinics are public or private? (medicines are free or with cost?)

Exclusion criteria: Please fix this phrase (not showing unwillingness)

Dosage individualization and optimization approaches:

Who did the interview for data collection in the hospital? External researchers or local healthcare practitioners? Please specify.

Who did the dose optimization? Pharmacists? Please describe them.

Did the researcher have collaboration with the prescribers (physicians)? Please clarify whether the individualization of the regimen was conducted with prescriber agreement?

Was there electronic health record in these 2 centers? Or used paper charts?

 

Data collection instrument, reliability and validity

Please specify which items have been evaluated using Cronbach alpha?

I cannot see scales (domains)n containing a group of items to measure the internal reliability?

Statistical analysis: Please specify the outcome variable and justify why authors used multinomial logistic regression over binary logistic regression?

How does this analysis help to answer the study objective?

Did you measure the medication adherence? since adherence is a high potential confounder can influence the renal disease progression.

Did you adjust (control) the effect of medication adherence on the renal disease progress? Since adherence is a high potential confounder can influence the regression outcome.

Results: Please specify the total number of patients in the beginning of the result section.

Page 6: How did you decide “need of dose individualization”? specify the measures/analysis used.

Limitations: Not measuring medication adherence can be a limitation since it is one of the study confounders.

Reviewer #6: This study would have add on literature if the authors employ intervention for all patients they have collected data from... and monitor them for a period of time to check medication benefit, compliance, uses, miss use. Communities need to be educated closely for such medications and doing clinical trials in a community level would be essential

6. PLOS authors have the option to publish the peer review history of their article (what does this mean?). If published, this will include your full peer review and any attached files.

Reviewer #1: No

Reviewer #2: No

Reviewer #3: No

Reviewer #4: No

Reviewer #5: **Yes: **Dr. Ali Azeez Al-Jumaili, BS Pharm, MS, MPH, PhD, University of Baghdad College of Pharmacy, Iraq and adjunct Assistant Professor at the University of Iowa College of Pharmacy, USA

Reviewer #6: **Yes: **Etwal Pierre Bou Raad

---

## [Author Response · Author response to Decision Letter 0]

16 Apr 2021

Thank you very much for your invaluable comments. We've addressed all the concerns point-by-point.

Responses to reviewers

Reviewer #1: The current bi-centric cross-sectional study conduct by Sapkota et al., has evaluated the dosage individualization proposed for anti-gout medications in the patients suffering from at two hospitals in Nepal. The authors have found that a total of 8% of the study participants needed dose individualization. This is an interesting topic, but the way the manuscript is presented makes it difficult for the readers to understand the findings of the study. Some of my comments to the authors are as follows:

Major comments

The authors have written down that “present study was aimed to individualize and optimize the dose of anti-gout medications among gout patients”. However their main findings are about frequency of patients who needed dose individualization. The aim of the study should redrafted.

Response: We've redrafted the aim as "The present study was probably the first of its kind in Nepal aiming to explore the status of individualization and optimization of the dose of anti-gout medications among gout patients."

The authors have used multinomial regression to find factors associated with the risk of progression of CKD in the study participants. First, this is not the major aim of the study and one gets confused that what is the need of this analysis in the current manuscript. Second, the results of multinomial regression have not been presented in the abstract. Third, progression or deterioration of a disease cannot be evaluated by a cross-sectional study design. What the authors have presented in the Results section of the main document are just the factors which have significant association with advanced stages of kidney diseases. It would have been good, if the authors have limited the results and discussion to the primary objective of the study. i.e. frequency of patients who needed dose individualization and the factors associated with it.

Response: First, we believed that exploration of risk factors or risk of progression of CKD would be useful while exploring the status of dosage individualization and optimization. Therefore, we applied multinomial regression for the same and we hope it would not deviate the main aims of the study. Second, we had already mentioned the information also in the Abstract section but had missed to specify, which we've now specified. 

Third, of course, we had also not tried to conclude the progression or deterioration of gout with the present cross-sectional study design, as this study design cannot do so. We just tried to document what appeared with the analysis. 

We thought that if we confine our results and discussion only to the frequency of patients who needed dose individualization and the factors associated with it, it would not depict the wholesome picture of the research. Therefore, we made an effort to go slightly beyond this. We're very sorry to bother you.

What criteria has been used for entering the independent variables into multinomial regression? 

Response: Independent variables (i.e., predictors) were entered into the multinomial regression based on evidence base and stepwise pattern by following the principle of parsimony. We've mentioned this in 'Statistical analysis' section in the manuscript.

Why 50% population proportion has been used in calculating the sample size for the current study? The authors should also give a bit detail about their random selection of the study subject, what type of random selection they have employed? How many patients or their records were approached, how many of them have been not included in the study and why?.

The value of “z” in the sample size calculation formula has been mistakenly written as 0.96, it should be written correct as 1.96.

Response: Since there was no exact data related to individualization and optimization of antigout medication dosage, we considered 50% population proportion while calculating the sample size for the current study. First, sample frame was created and then the study subjects were randomly selected by applying lottery method. All patients taking anti-gout medications along with laboratory reports consisting of creatinine value were included and those not taking anti-gout medications and not showing unwillingness to participate in the research were excluded. We are sorry for typo error in the value of 'z' and we've corrected this now.

In the methods section of the paper it has been written as “…..study was carried out among new gout patients visiting two gout treatment centers”, whereas in the Inclusion, exclusion part of the paper it has been written as “……All patients taking anti-gout medications along with laboratory reports consisting of creatinine value were included for the research”. As the authors have included established gout patients who were taking anti-gout medicines, the authors should make it clear that they mean by “new gout patients”

Response: We've omitted the confusing term i.e., 'new' from the Methods section and we've further elaborated the inclusion criteria to avoid the confusion of 'new patients'. Actually, those patients with gout who came to visit the study centers and who were previously (since last six months from the date of data collection) or currently taking antigout medications were enrolled for the study.

In the results sections the authors have mentioned that “….The multinomial logistic regression analysis showed that moderate risk of progression to CKD decreased with advancing age but that of high risk increased in the age range 54-63 and ≥84 years by 17.77 and 43.02 times, respectively”. What the authors meant by moderate risk and high risk? What criteria has been used for categorizing the risk of renal disease progression as moderate and high?

Response: The risk of progression (moderate or high) was determined by using the National Kidney Foundation (NKF) GFR Calculator (https://www.kidney.org/professionals/kdoqi/gfr_calculator). We had already mentioned this in the 'Dosage individualization and optimization approaches' section.

Overall the discussion is too vague and does not revolve around the main findings of the study.

Response: We've removed the vague statements and abridged the Discussion section to make it compatible with the main findings of the study and critical review with other literatures.

There are many typo and grammatical mistakes which need correction.

Response: We've thoroughly gone through the typo and grammatical mistakes and corrected the same.

Reviewer #2: Dear Editor

The following are my observations on the article “Dosage individualization proposed for anti-gout medications among the patients with gout: a bicentric study in Nepal”.

1. The inclusion and the exclusion criteria to be more specific:

Example: The patients taking anti gout medications are recruited in the study, the authors does not mention, how long the patients on medications can be recruited. Based on the duration of treatment, the outcome of the assessment varies. Similarly, the exclusion criteria need to be specific.

Response: We've now further elaborated upon the inclusion and exclusion criteria. We hope this will find well.

2. In the discussion section, most of the data mentions about the results. Discussion should be re-written.

Response: We've rewritten Discussion section by removing many of the unnecessary statements.

Reviewer #3: Comment 1: Typographical error was observed while calculating sample size.

Response: We've corrected the typographical error.

Comment 2: Grammatical errors were observed in exclusion criteria, it is giving different meaning, please rewrite the sentence.

Response: Thank you very much for indicating our mistake. We've corrected the term 'unwillingness', which was incorrectly written.

Comment 3: Abbreviations should be defined at first mention and used consistently thereafter.

Response: We’ve gone through the whole manuscript and adjusted accordingly.

Comment 4: The word country-wide supposed to be country-wise in line 1 and 2 at page 3. Kindly check it.

Response: We've corrected accordingly.

Comment 5: Please re-write the line number 3 and 4 at page number 3.

Response: We've rewritten the same.

Comment 6: In data section, can you elaborate the tools used for assessment, or the questionnaire used and the value of Cronbach analysis

Response: We've elaborated upon the references on which we based our study tool in the 'Data collection instrument, reliability and validity' section. However, we had developed our own study tool, taking references mainly from these. Value of Cronbach analysis was 0.707.

Comment 7: In the conclusion section, can you elaborate that which of the 30 individuals require the individualization.

Response: Dose individualization was felt necessary for those patients taking colchicine 500 mcg, etoricoxib 90 mg, febuxostat 40 mg and 80 mg, methylprednisolone inj. 80 mg, prednisolone 2.5 mg, 5 mg, 15 mg 20 mg and 30 mg, and prednisolone 5/2.5 mg. Altogether the total number of cases was 30, which was already depicted in Table 3 as well.

Overall, this manuscript needs major revision including write-up and supportive evidence.

Response: We've thoroughly revised our manuscript, especially in Methods and Discussion sections.

Reviewer #4: Comments

1. Please describe SRS procedure.

Response: We've elaborated upon the SRS technique.

2. Researcher perform any pilot study? If yes then please describe. If not how it could be fulfill face validation? Which is required for any developed questionnaire.

Response: Yes, of course we performed pilot study among 10% of the projected sample size (i.e., 39).

3. Better to report the content validity results with CVR and CVI.

Response: We are very sorry not to report the content validity results with CVR and CVI as our expert panel did not suggest us so. We'll definitely address such concern in future research.

4. Outcome variable is not describe properly.

Response: Outcome variable was risk of progression to CKD (with reference to low risk) and such risk was determined by using the National Kidney Foundation (NKF) GFR Calculator (https://www.kidney.org/professionals/kdoqi/gfr_calculator).

5. What was the variable selection procedure for Multinomial regression model?

Response: We've elaborated upon the variable selection procedure for Multinomial regression model in the 'Statistical analysis' section.

6. Imaginary results observed from Multinomial regression especially from OR and CI. Due to inappropriate grouping of predictors (lot of groups included zero frequency) as well as inappropriate reference category inflation happened on OR and CI.

Response: We exactly kept the same OR and CI which was obtained from the multinomial logistic regression analysis. Yes, of course there were lots of groups with zero frequency but in different risk categories. Therefore, we thought it would be appropriate to show exactly the frequencies. Now, we've replaced zero frequency with '-' sign.

7. Suggesting to reanalyze data after performing proper categorization among predictors with bivariate analysis to select predictors for multiple model.

Response: Since the present multinomial logistic regression was based on the predictors selected from the literature base, it was not possible to categorize these into bivariate categories. And also since multinomial logistic regression analysis was meant for multiple predictors, we are extremely sorry to stick on the current analysis. 

Reviewer #5: Dosage individualization proposed for anti-gout medications among the patients with gout: A bicentric study in Nepal

PONE-D-20-38874

Thank you very much for the opportunity to review this manuscript. I invite authors to resubmit your manuscript after addressing all comments. Please carefully consider all issues I mentioned, outline every change made point by point and send us 2 documents: revised manuscript with track changes and author responses to the reviewer comments. Thanks

Response: Thank you very much for your expert reviews. We've addressed all your comments pont-by-point and done accordingly as you suggested.

Abstract:

Objective: Was the objective to optimize the therapy or prove the necessity of the dose optimization to minimize the predicted progression to chronic kidney disease (CKD) ? As the conclusion shows the need (thirty (8%) cases required dose individualization) rather than outcome of the individualization.

Response: The objective was to explore the necessity of the dose optimization to minimize the predicted progression to chronic kidney disease (CKD). We've clarified this in the revised manuscript. We've clarified this on the Conclusion section as well.

Methods: Bicentric cross-sectional” The term bicentric can be deleted since it is not part of the study design (cross-sectional) while it is part of the study setting (in 2 centers).

Response: We've deleted the term 'bicentric' from the Methods section.

Not showing unwillingness. Please do not use double negative, so reword to either “not showing willingness” or showing unwillingness.

Response: Thank you very much for indicating our typo error. We’ve corrected this in the revised manuscript.

I am not sure how cross-sectional design can prove medicine individualization? I mean without prospective follow-up of the cases how can the authors prove the effectiveness of the individualized regimen?

Response: Yes, of course cross-sectional design cannot prove medicine individualization as it has its inherent limitations and we need longitudinal studies or clinical trials for the same. We've just depicted what we observed from the analysis of the predictors and outcome variables during the study period, with the help of statistical analysis.

Methods

Study design, study area and study population

Please specify whether those 2 outpatient clinics are public or private? (medicines are free or with cost?)

Response: Both of the study centers were privately owned and patients had to pay for the medicine, laboratory, radiologica and other health services out of their own pocket. We've also mentioned the statement in 'Study design, study area and study population' heading.

Exclusion criteria: Please fix this phrase (not showing unwillingness)

Response: We've corrected the typo error.

Dosage individualization and optimization approaches:

Who did the interview for data collection in the hospital? External researchers or local healthcare practitioners? Please specify.

Response: Data collection from the two study centers was done by four researchers, except the principal investigator, in collaboration with the practicing physicians. We've mentioned this in 'Data collection instrument, reliability and validity' section.

Who did the dose optimization? Pharmacists? Please describe them.

Response: Yes, of course need of the dose individualization was raised by the pharmacists in collaboration with the physicians. We've mentioned this in 'Data collection instrument, reliability and validity' section.

Did the researcher have collaboration with the prescribers (physicians)? Please clarify whether the individualization of the regimen was conducted with prescriber agreement?

Response: Yes, of course need of the dose individualization was raised by the pharmacists in collaboration with the physicians. We've mentioned this in 'Data collection instrument, reliability and validity' section.

Was there electronic health record in these 2 centers? Or used paper charts?

Response: Both of the centers were using paper prescriptions at the time of data collection.

Data collection instrument, reliability and validity

Please specify which items have been evaluated using Cronbach alpha? I cannot see scales (domains)n containing a group of items to measure the internal reliability?

Response: Yes, of course scale variables or items were evaluated using Cronbach alpha and since dose individualization related items were not scale, these were beyond the scope of Cronbach alpha computation.

Statistical analysis: Please specify the outcome variable and justify why authors used multinomial logistic regression over binary logistic regression?

How does this analysis help to answer the study objective?

Response: Outcome variable was risk of progression to CKD (with reference to low risk) and such risk was determined by using the National Kidney Foundation (NKF) GFR Calculator (https://www.kidney.org/professionals/kdoqi/gfr_calculator). Since the present multinomial logistic regression was based on the predictors selected from the literature base, it was not possible to categorize these into bivariate categories. And also since multinomial logistic regression analysis was meant for multiple predictors, and this helped explore association between the predictors and outcome variables to add values to the study objective, we chose the current analysis. 

Did you measure the medication adherence? since adherence is a high potential confounder can influence the renal disease progression.

Response: We are sorry we did not measure the medication adherence.

Did you adjust (control) the effect of medication adherence on the renal disease progress? Since adherence is a high potential confounder can influence the regression outcome.

Response: We are sorry we did not adjust (control) the effect of medication adherence on the renal disease progress.

Results: Please specify the total number of patients in the beginning of the result section.

Response: We have indicated this in the result section.

Page 6: How did you decide “need of dose individualization”? specify the measures/analysis used.

Response: We had already mentioned the 'Dosage individualization and optimization approaches' and doses to be individualized were decided based on standard references of the Renal Drug Handbook 4th edition and verified with the British National Formulary 80. Dose to be individualized was later discussed with the prescribing physicians.

Limitations: Not measuring medication adherence can be a limitation since it is one of the study confounders.

Response: We've indicated the fact in the 'Limitations' section.

Reviewer #6: This study would have add on literature if the authors employ intervention for all patients they have collected data from... and monitor them for a period of time to check medication benefit, compliance, uses, miss use. Communities need to be educated closely for such medications and doing clinical trials in a community level would be essential

Response: Thank you very much for your valuable suggestions. We've planned to conduct intervention for the patients in future, monitor them for a period of time to check medication benefit, compliance, uses, misuse and other relevant issues. We've also planned to educate the community people but in future separate project.

---

## [Decision Letter · Decision Letter 1]

7 Jun 2021

PONE-D-20-38874R1

Dosage individualization proposed for anti-gout medications among the patients with gout: A bicentric study in Nepal

PLOS ONE

Dear Dr. Sapkota,

Thank you for submitting your manuscript to PLOS ONE. After careful consideration, we feel that it has merit but does not fully meet PLOS ONE’s publication criteria as it currently stands. Therefore, we invite you to submit a revised version of the manuscript that addresses the points raised during the review process.

We look forward to receiving your revised manuscript.

Kind regards,

Tauqeer Hussain Mallhi, Ph.D

Academic Editor

PLOS ONE

Additional Editor Comments (if provided):

Thank you for submitting the revise draft. Though major revisions have been addressed by the authors but still manuscript requires attention to improve the grammar, syntax and scientific writing. I invite authors to respond the comments from the reviewer so an appropriate decision can be taken.

Reviewers' comments:

Reviewer's Responses to Questions

**Comments to the Author**

1. If the authors have adequately addressed your comments raised in a previous round of review and you feel that this manuscript is now acceptable for publication, you may indicate that here to bypass the “Comments to the Author” section, enter your conflict of interest statement in the “Confidential to Editor” section, and submit your "Accept" recommendation.

Reviewer #2: All comments have been addressed

Reviewer #3: (No Response)

Reviewer #5: All comments have been addressed

2. Is the manuscript technically sound, and do the data support the conclusions?

Reviewer #2: Yes

Reviewer #3: Yes

Reviewer #5: Yes

3. Has the statistical analysis been performed appropriately and rigorously? 

Reviewer #2: Yes

Reviewer #3: Yes

Reviewer #5: Yes

4. Have the authors made all data underlying the findings in their manuscript fully available?

Reviewer #2: Yes

Reviewer #3: Yes

Reviewer #5: (No Response)

5. Is the manuscript presented in an intelligible fashion and written in standard English?

Reviewer #2: Yes

Reviewer #3: No

Reviewer #5: Yes

6. Review Comments to the Author

Reviewer #2: All the responses have been adequately answered by the Author. The authors have in detail mentioned the inclusion and exclusion criteria of the study. The write up of the discussion is rectified.

Reviewer #3: The manuscript describes Dosage individualization proposed for anti-gout medications among the patients with gout: A bicentric study in Nepal. Overall, this manuscript contains lots of grammatical and typographical errors, suggested to use more presentable language apart from this manuscript contains valuable information which may be useful in future intervention studies.

Please find the other comments in the attached file.

Reviewer #5: Thank you addressing my comments, but please make sure to include all the responses/corresponding revisions in your manuscript as changes.

7. PLOS authors have the option to publish the peer review history of their article (what does this mean?). If published, this will include your full peer review and any attached files.

Reviewer #2: No

Reviewer #3: No

Reviewer #5: **Yes: **Dr. Ali Azeez Al-Jumaili, PhD

---

## [Author Response · Author response to Decision Letter 1]

14 Jun 2021

Response to reviewer's comments

We are very much thankful to the editor and the reviewers for their valuable suggestions for our manuscript. We have tried our best to address all the concerns point-by-point. 

Reviewer #3: The manuscript describes Dosage individualization proposed for anti-gout medications among the patients with gout: A bicentric study in Nepal. Overall, this manuscript contains lots of grammatical and typographical errors, suggested to use more presentable language apart from this manuscript contains valuable information which may be useful in future intervention studies.

Please find the other comments in the attached file.

Response: We have tried to remove grammatical and typographical errors by thoroughly revising the manuscript again. We have also tabulated the responses as below: 

Page &Line No. Comments Response

Abstract 

Page 1& line No. 23 Kindly replace unwillingness with willingness in sentence Changed

Page 2& line No. 1- 2 Kindly re-write this sentence, as meaning is not clear. Rewritten

Methods 

Page 4& Line No. 5 and 7 Kindly correct the spelling of “radiological” and “first clinic” Corrected

Page 5 & Line No. 21 Kindly correct the typographical error, instead of “CDC guidance”write“CDC guidelines”. Corrected

Page 5& Line No. 27 Suggested to elaborate or outline the structure of self-developed tool used for data collection. Structure of the self-developed tool elaborated

Results 

Page 6 Suggested to use either frequency or percentage to define the findings of variables in the manuscript. Only percentage kept and frequency removed.

Discussion 

Page 7& Line No. 27-28 Kindly reform this sentence. Reformed as suggested

Other Limitation of this study can critically affect the reproducibility of this study (non-adherence must be checked to maintain reproducibility) Measuring adherence is as such a tricky part in therapy and this might be under the scope of another separate research.

Reviewer #2: All the responses have been adequately answered by the Author. The authors have in detail mentioned the inclusion and exclusion criteria of the study. The write up of the discussion is rectified.

Responses: Thank you very much. We have again thoroughly revised the manuscript.

Reviewer #5: Thank you addressing my comments, but please make sure to include all the responses/corresponding revisions in your manuscript as changes.

Response: Thank you very much. We have also included all the responses in the revised manuscript.

Additional Editor Comments (if provided):

Thank you for submitting the revise draft. Though major revisions have been addressed by the authors but still manuscript requires attention to improve the grammar, syntax and scientific writing. I invite authors to respond the comments from the reviewer so an appropriate decision can be taken.

Response: We have again thoroughly revised the manuscript on this second revision as well. We hope the manuscript has now been improved in terms of the grammar, syntax and scientific writing. We have also abridged the title, retaining the original meaning.

---

## [Decision Letter · Decision Letter 2]

1 Jul 2021

PONE-D-20-38874R2

Dosage individualization proposed for anti-gout medications among the patients with gout

PLOS ONE

Dear Dr. Sapkota,

Thank you for submitting your manuscript to PLOS ONE. After careful consideration, we feel that it has merit but does not fully meet PLOS ONE’s publication criteria as it currently stands. Therefore, we invite you to submit a revised version of the manuscript that addresses the points raised during the review process.

We look forward to receiving your revised manuscript.

Kind regards,

Tauqeer Hussain Mallhi, Ph.D

Academic Editor

PLOS ONE

Journal Requirements:

Reviewers' comments:

Reviewer's Responses to Questions

**Comments to the Author**

1. If the authors have adequately addressed your comments raised in a previous round of review and you feel that this manuscript is now acceptable for publication, you may indicate that here to bypass the “Comments to the Author” section, enter your conflict of interest statement in the “Confidential to Editor” section, and submit your "Accept" recommendation.

Reviewer #2: All comments have been addressed

Reviewer #3: All comments have been addressed

2. Is the manuscript technically sound, and do the data support the conclusions?

Reviewer #2: Yes

Reviewer #3: Yes

3. Has the statistical analysis been performed appropriately and rigorously? 

Reviewer #2: Yes

Reviewer #3: Yes

4. Have the authors made all data underlying the findings in their manuscript fully available?

Reviewer #2: Yes

Reviewer #3: Yes

5. Is the manuscript presented in an intelligible fashion and written in standard English?

Reviewer #2: Yes

Reviewer #3: No

6. Review Comments to the Author

Reviewer #2: All the comments have been satisfactorily addressed. No further comments. The manuscript can be accepted.

Reviewer #3: All the comments have been address adequately by author. However, few more comments need to be address.

Comment 1 How the study will change the current prescribing pattern of anti-gout therapy with respect to dosage individualization (include in discussion) as the drugs were prescribed generally according to the treatment guidelines.

Comment 2 In this study, patients diagnosed with gout were included, Kindly include the time period of therapy which they are taking like from one year or two year or naive patients (in tabular format in results)

Comment 3 Some of the patient were presented with their complications, was their therapy regimen was taken care of, like w.r.t drug interactions and other body parameters, if yes please include the status in results section.

Comment 4 On an average, how many follow-ups were taken w.r.t. each patient. Kindly include a rough average

Comment 5 What was the dosage regimen of corticosteroids of patients with complications if it was tapered or escalated what pattern was used? Please elaborate

7. PLOS authors have the option to publish the peer review history of their article (what does this mean?). If published, this will include your full peer review and any attached files.

Reviewer #2: No

Reviewer #3: No

---

## [Author Response · Author response to Decision Letter 2]

1 Jul 2021

Response to reviewer's comments

We are very much thankful to the editor and the reviewers for their valuable suggestions for our manuscript. We've tried our best to address all the concerns point-by-point and have also revised the manuscript in light of these, where necessary. 

Reviewer #2: All the comments have been satisfactorily addressed. No further comments. The manuscript can be accepted.

Response: Thank you very much.

Reviewer #3: All the comments have been address adequately by author. However, few more comments need to be address.

Comment 1 How the study will change the current prescribing pattern of anti-gout therapy with respect to dosage individualization (include in discussion) as the drugs were prescribed generally according to the treatment guidelines.

Response: We've kept the following statement at the bottom of the Discussion section.

The study findings may help the practitioners select the optimum regimen of anti-gout therapy for the individual patient based on the dosage individualization proposed.

Comment 2 In this study, patients diagnosed with gout were included, Kindly include the time period of therapy which they are taking like from one year or two year or naive patients (in tabular format in results).

Response: We've indicated the duration of disease in tables 1 and 2, with clear units in years. The patients were on antigout therapy since they were diagnosed, but we followed the inclusion criteria (which we had already mentioned in our inclusion criteria) as: 

Those patients who were freshly prescribed with antigout medications and/or those taking the same since last six months (from the date of data collection) were enrolled.

Comment 3 Some of the patient were presented with their complications, was their therapy regimen was taken care of, like w.r.t drug interactions and other body parameters, if yes please include the status in results section.

Response: The report was discussed with the prescribing physicians, and rest of the therapeutic decision making was left to them. The outcome was not followed up in the present research, as it was a cross-sectional study.

Comment 4 On an average, how many follow-ups were taken w.r.t. each patient. Kindly include a rough average.

Response: Since the study was cross-sectional, no follow up of the patient was performed within the study period. However, the prescribing physicians asked their patients to come up on follow up every six month.

Comment 5 What was the dosage regimen of corticosteroids of patients with complications if it was tapered or escalated what pattern was used? Please elaborate.

Response: Since there was no such report of complications with the dosage regimen of corticosteroids within the study period, this was not worrying case here.

Other revisions performed:

Tables 1 and 2: We've indicated the duration of disease in years, which was previously missing in table.

Table 2: We've shifted the descriptions of obesity status in the footnote to minimize the table length.

---

## [Decision Letter · Decision Letter 3]

28 Jul 2021

PONE-D-20-38874R3

Dosage individualization proposed for anti-gout medications among the patients with gout

PLOS ONE

Dear Dr. Sapkota,

Thank you for submitting your manuscript to PLOS ONE. After careful consideration, we feel that it has merit but does not fully meet PLOS ONE’s publication criteria as it currently stands. Therefore, we invite you to submit a revised version of the manuscript that addresses the points raised during the review process.

We look forward to receiving your revised manuscript.

Kind regards,

Tauqeer Hussain Mallhi, Ph.D

Academic Editor

PLOS ONE

Journal Requirements:

Additional Editor Comments (if provided):

Thank you for revising the draft. This manuscript is improved but still needs to address few concerns raised by the reviewers. Please revise the draft accordingly and submit at your convenience.

Reviewers' comments:

Reviewer's Responses to Questions

**Comments to the Author**

1. If the authors have adequately addressed your comments raised in a previous round of review and you feel that this manuscript is now acceptable for publication, you may indicate that here to bypass the “Comments to the Author” section, enter your conflict of interest statement in the “Confidential to Editor” section, and submit your "Accept" recommendation.

Reviewer #3: All comments have been addressed

2. Is the manuscript technically sound, and do the data support the conclusions?

Reviewer #3: Yes

3. Has the statistical analysis been performed appropriately and rigorously? 

Reviewer #3: Yes

4. Have the authors made all data underlying the findings in their manuscript fully available?

Reviewer #3: No

5. Is the manuscript presented in an intelligible fashion and written in standard English?

Reviewer #3: Yes

6. Review Comments to the Author

Reviewer #3: The data of co-morbidities in the manuscript should be clearly stated, they can be summarized in tabular form for a clear representation of the result. Table 1 and 2 are suggested to be reframed in accordance to the correct scientific format.

7. PLOS authors have the option to publish the peer review history of their article (what does this mean?). If published, this will include your full peer review and any attached files.

Reviewer #3: No

---

## [Author Response · Author response to Decision Letter 3]

28 Jul 2021

Reviewers' comments:

Reviewer's Responses to Questions

Comments to the Author

1. If the authors have adequately addressed your comments raised in a previous round of review and you feel that this manuscript is now acceptable for publication, you may indicate that here to bypass the “Comments to the Author” section, enter your conflict of interest statement in the “Confidential to Editor” section, and submit your "Accept" recommendation.

Reviewer #3: All comments have been addressed

Response: Thank you very much for your confirmation.

2. Is the manuscript technically sound, and do the data support the conclusions?

Reviewer #3: Yes

Response: Thank you very much for your confirmation.

3. Has the statistical analysis been performed appropriately and rigorously?

Reviewer #3: Yes

Response: Thank you very much for your confirmation.

4. Have the authors made all data underlying the findings in their manuscript fully available?

Reviewer #3: No

Response: Full data would be made available upon request to the corresponding author. Data from individual patient would make the journal's space unnecessary space and we think these will not be of interest and importance to all the prospective researchers. The required information has already been supplied in S1 and S2 appendices.

5. Is the manuscript presented in an intelligible fashion and written in standard English?

Reviewer #3: Yes

Response: Thank you very much for your confirmation.

6. Review Comments to the Author

Reviewer #3: The data of co-morbidities in the manuscript should be clearly stated, they can be summarized in tabular form for a clear representation of the result. Table 1 and 2 are suggested to be reframed in accordance to the correct scientific format.

Response: The data of co-morbidities were already displayed at the 'S1 Appendix: Comorbidities of the patients'. Table 1 and 2, and S1 and S2 appendices have now been reframed in accordance to the correct scientific format.

---

## [Decision Letter · Decision Letter 4]

24 Aug 2021

Dosage individualization proposed for anti-gout medications among the patients with gout

PONE-D-20-38874R4

Dear Dr. Sapkota,

We’re pleased to inform you that your manuscript has been judged scientifically suitable for publication and will be formally accepted for publication once it meets all outstanding technical requirements.

Kind regards,

Tauqeer Hussain Mallhi, Ph.D

Academic Editor

PLOS ONE

Additional Editor Comments (optional):

Reviewers' comments:

Reviewer's Responses to Questions

**Comments to the Author**

1. If the authors have adequately addressed your comments raised in a previous round of review and you feel that this manuscript is now acceptable for publication, you may indicate that here to bypass the “Comments to the Author” section, enter your conflict of interest statement in the “Confidential to Editor” section, and submit your "Accept" recommendation.

Reviewer #3: All comments have been addressed

2. Is the manuscript technically sound, and do the data support the conclusions?

Reviewer #3: Yes

3. Has the statistical analysis been performed appropriately and rigorously? 

Reviewer #3: Yes

4. Have the authors made all data underlying the findings in their manuscript fully available?

Reviewer #3: Yes

5. Is the manuscript presented in an intelligible fashion and written in standard English?

Reviewer #3: Yes

6. Review Comments to the Author

Reviewer #3: Manuscript write up is up to the mark and the author had pulled out all infirmity also he had delectably fixed all suggestions prescribed by the reviewer. The introduction is relevant and theory based. The language used is appropriate and scientific.

7. PLOS authors have the option to publish the peer review history of their article (what does this mean?). If published, this will include your full peer review and any attached files.

Reviewer #3: No

---

## [Editor Report · Acceptance letter]

3 Sep 2021

PONE-D-20-38874R4 

Dosage individualization proposed for anti-gout medications among the patients with gout 

Dear Dr. Sapkota:

I'm pleased to inform you that your manuscript has been deemed suitable for publication in PLOS ONE. Congratulations! Your manuscript is now with our production department. 

Kind regards, 

on behalf of

Dr. Tauqeer Hussain Mallhi 

Academic Editor

PLOS ONE